# Renoprotective effects of paramylon, a β-1,3-D-Glucan isolated from *Euglena gracilis* Z in a rodent model of chronic kidney disease

Yoshikuni Nagayama[1]*, Naoyuki Isoo[1], Ayaka Nakashima[2], Kengo Suzuki[2], Mizuki Yamano[1], Tomoyuki Nariyama[1], Motoka Yagame[1], Katsuyuki Matsui[1]

1 Department of Internal Medicine IV, Teikyo University School of Medicine, University Hospital, Mizonokuchi, Kawasaki, Japan, 2 Euglena Co., Ltd., Tokyo, Japan

* ynaga@med.teikyo-u.ac.jp

## Abstract

Paramylon is a novel β-glucan that is stored by *Euglena gracilis Z*, which is a unicellular photosynthesizing green alga with characteristics of both animals and plants. Recent studies have indicated that paramylon functions as an immunomodulator or a dietary fiber. Currently, chronic kidney disease (CKD) is a global health problem, and there is no effective preventive treatment for CKD progression. However, paramylon may suppress the progression of CKD via the elimination of uremic toxins or modulation of gut microbiota, leading to the alleviation of inflammation. The aim of this study was to evaluate the effect of paramylon in CKD rat model. Eight-week-old male Wistar rats with a 5/6 nephrectomy were given either a normal diet or a diet containing 5% paramylon for 8 weeks. Proteinuria was measured intermittently. Serum and kidney tissues were harvested after sacrifice. We performed a renal molecular and histopathological investigation, serum metabolome analysis, and gut microbiome analysis. The results showed that paramylon attenuated renal function, glomerulosclerosis, tubulointerstitial injury, and podocyte injury in the CKD rat model. Renal fibrosis, tubulointerstitial inflammatory cell infiltration, and proinflammatory cytokine gene expression levels tended to be suppressed with paramylon treatment. Further, paramylon inhibited the accumulation of uremic toxins, including tricarboxylic acid (TCA) cycle-related metabolites and modulated a part of CKD-related gut microbiota in the CKD rat model. In conclusion, we suggest that paramylon mainly inhibited the absorption of non-microbiota-derived uremic solutes, leading to protect renal injury via anti-inflammatory and anti-fibrotic effects. Paramylon may be a novel compound that can act against CKD progression.

## Introduction

β-glucans are types of polysaccharides that consist of D-glucose units linked via β-glycosidic bonds. They are present in various foods such as cereals, mushrooms, yeasts, and seaweeds [1]. The biological effects of these dietary β-glucans include immunostimulation [2], immune regulation of the T helper 1 (Th1)/Th2 balance [3], cholesterol-lowering property [4], antioxidant

**Data Availability Statement:** All relevant data are within the manuscript and its Supporting Information files.

**Funding:** Ms. Ayaka Nakashima and Dr. Kengo Suzuki are employees of euglena Co., Ltd. They adjusted the test concept, created a sample for this experiment (preparing resources), and discussed the results obtained. This research did not receive any specific grant from funding agencies in the public, commercial, or not-for-profit sectors.

**Competing interests:** Ms. Ayaka Nakashima and Dr. Kengo Suzuki are employees of euglena Co., Ltd. There are no patents, products in development or marketed products to declare. This does not alter the authors' adherence to all the PLoS ONE policies on sharing data and materials.

activity [5], and antitumor activity [6–8]. Paramylon is a novel β-glucan that is stored by *Euglena gracilis Z*(*E. gracilis Z*), which is a unicellular photosynthesizing green alga with characteristics of both animals and plants. It is a natural substance and represents 30–70% of the dry weight of *E. gracilis Z*. Like other types of β-glucans, paramylon also has been shown to have immunostimulation [9], antioxidant activity [10], and immunomodulating effects including anti-atopic dermatitis [11], antihuman immunodeficiency virus activity [12], antimicrobial activity [13], anti-influenza virus activity [14], and suppressive Th17 immunity activity [15]. Additionally, paramylon is a dietary fiber, and a *Euglena*-diet containing paramylon has suppressed cholesterol and fat absorption in the digestive tract [16].

Chronic kidney disease (CKD) is a global health problem that is associated with high risk for cardiovascular morbidity and death [17, 18]. Because the precise mechanisms for CKD progression remain unclear, the main strategy to prevent deterioration of kidney function is to resolve the risk factors, including hypertension, hyperglycemia, hyperlipidemia, proteinuria, smoking, and obesity. Moreover, as CKD progresses, the accumulation of uremic toxins in CKD patients causes renal injury and fibrosis [19]. Recently, it was reported that CKD profoundly changes the intestinal microbial flora [20], leading to a suggestion that intervention for the gut microbiota derangements with probiotics and/or dietary fibers may be useful for decreasing the uremic toxin production and improve renal function [21, 22].

Paramylon, as an immunomodulator and a dietary fiber, may suppress the progression of CKD by eliminating uremic toxins or modulating gut microbiota, thereby, alleviating inflammation. However, the effects of paramylon in CKD remains unknown. The aim of this study was to investigate the effect of paramylon in CKD rat model.

## Materials and methods

### Test substance

*E. gracilis Z* was a powder product, and the nutritional analysis results, which were similar to those of a previous report [23] are as follows: carbohydrates 29.4%, protein 42.3%, and lipid 19.0%. Approximately 70–80% of the carbohydrate content was paramylon, which was isolated from *E. gracilis Z* obtained from the euglena Co., Ltd., (Tokyo, Japan). The following usual method of preparation of paramylon was used. Cultured *E. gracilis Z* cells were collected by continuous centrifugation and washed with water. After suspending in water, the cells were broken down using ultrasonic waves and the contents, which contained paramylon, was collected. To remove the lipid and protein, the crude paramylon preparation was treated with 1% sodium dodecyl sulfate (SDS) solution at 95°C for 1 h, and then at 50°C for 30 min with 0.1% SDS. After centrifugation, the paramylon was obtained, and further refined by repeated washing with water, acetone, and ether, sequentially. *E. gracilis* Z containing paramylon has been confirmed to be safe in animal tests and known that heavy metals and pesticides are not detected. The in-house lot number for paramylon administered to animals was PA121025. A special pellet diet containing 5% w/w pure paramylon was prepared (euglena Co., Ltd, Tokyo, Japan) and administered to the animals.

### Animal experiments

Eight-week-old male Wistar rats (Sankyo Labo Service, Tokyo, Japan) were used. The rats were housed in a breeding cage in groups in an animal room maintained at a temperature of 24±2°C, relative humidity of 50±5%, full ventilation, and illumination at 7:00–21:00. The rats had access to drinking water and chow *ad libitum*. The rats were acclimatized for one week. Then, the rats received a surgical resection of the upper and lower one-third of the left kidney or a sham operation of the left kidney. Two weeks later, the rats received a right

uninephrectomy or a sham operation of the right kidney. All surgery was performed under isoflurane inhalation anesthesia. Then, 5/6 nephrectomy (Nx) rats were randomized into two groups, i.e., a normal diet group and the paramylon treatment group (Nx + PAR). A normal pellet diet (Labo MR Stock, NOSAN Japan, Inc) was given to the sham controls (n = 4) and the Nx rats (n = 8) for 8 weeks. A special pellet containing 5% w/w pure paramylon in the normal pellet (euglena Co., Ltd, Tokyo, Japan) was given for 8 weeks to Nx + PAR rats (n = 8). At the end of the experiment, the rats were euthanized under analgesia with medetomidine hydrochloride, midazolam, and butorphanol, after which blood and tissue samples were obtained. Three days before sacrificing, rats were placed in an individual metabolic cage to measure water and food intake. One day before sacrificing, rats were placed in an individual metabolic cage to collect urine and feces for 12 hours after overnight fasting. Animal experiments were approved by the Teikyo University Ethics Committee for Animal Experiments (#15–044) and were conducted in accordance with the guidelines of the Institute Animal Care and Use Committee of the Teikyo University.

## Serum and urine measurements

Urinary total protein and creatinine levels, serum creatinine, total cholesterol, and urea nitrogen levels were measured by routine methods at BML, INC. (Tokyo, Japan).

## Kidney histology, immunostaining, and quantification

Kidney sections were processed as described previously [24]. Slides for routine morphology were stained with periodic acid-Schiff (PAS) and Masson's trichrome. The primary antibodies that were used are listed in S1 Table. For transmission electron microscopy (TEM), the kidneys were cut into 1–2 mm$^3$ pieces. The tissues were prefixed immediately with 2.5% glutaraldehyde in 0.1 M phosphate buffer (pH 7.4) for 48–96 h and postfixed with buffered 1% osmium tetroxide for 90 minutes. The fixed tissues were dehydrated with ethanol and embedded in Quetol 812 mixture (Nisshin EM, Tokyo, Japan). Ultrathin 80 nm sections were cut with an EM UC7 Ultramicrotome (Leica, Wetzlar, Germany) and stained with 4% uranyl acetate and Sato's lead staining solutions [25]. The sections were observed with a HT7700 TEM (Hitachi, Tokyo, Japan) at 80 kV. The pictures of the podocyte foot processes were taken at a magnification of 5000x.

Sixty to 100 PAS-stained glomeruli for each animal were evaluated for glomerular sclerosis. The glomerular sclerotic score was graded on a scale of 0 to 4, as described previously [26]. In short, glomerular sclerosis was assessed with a semiquantitative score, i.e., 0 = absent; 1 = 1% to 25%; 2 = 26% to 50%; 3 = 51% to 75%; and 4 = 76% to 100% of the glomerular tuft area. Glomerular sclerosis was defined as the disappearance of cellular elements from the tuft, capillary loop collapse, and folding of the glomerular basement membrane (GBM) with accumulation of amorphous material. Forty PAS-stained renal cortices in a high-power field for each animal were evaluated for tubular injury. The tubular injury score was graded on a scale of 0 to 4, as described previously [27]. In short, tubular injury was assessed with a semiquantitative score, i.e., 0 = absent; 1 = 1% to 10%; 2 = 11% to 25%; 3 = 26% to 75%; and 4 = 76% to 100% of the tubular area. Tubular injury was defined as tubular epithelial necrosis, tubular atrophy, tubular dilatation, or thickening of the tubular basement membrane. Fifty PAS-stained glomeruli for each animal were evaluated for the glomerular cross-sectional area using ImageJ software version 1.45. The fibrotic area according to Masson's trichrome staining or α-smooth muscle actin (α-SMA) expression in 20 renal cortices for each animal was quantified using ImageJ software version 1.45 in a high-power field. The area of positive staining was calculated as the percentage of the total area, as described previously [28]. Monocytes/macrophages (ED-1),

CD3, or proliferating cell nuclear antigen (PCNA)-positive cells were counted in 20 renal cortices for each animal in a high-power field. For quantitative ultrastructural analysis of the glomerulus by TEM, the number of podocyte foot processes present in each micrograph was divided by the total length of the GBM region in each image to decide the mean density of podocyte foot processes, as described previously [29]. The length of the GBM was measured using ImageJ software version 1.45. These microscopic analyses were performed in a blinded fashion.

## Quantitative real-time Reverse Transcriptase Polymerase Chain Reaction (RT-PCR) analysis

Total RNA from whole kidney was extracted using a RNeasy Lipid Tissue Mini Kit (Qiagen, Hilden, Germany). cDNA synthesis was performed using a SuperScript VILO™ cDNA Synthesis Kit (Invitrogen, Carlsbad, CA). Quantitative real-time RT-PCR analysis was performed using a TaqMan Gene Expression Assay (Thermo Fisher Scientific, Waltham, MA). Quantitative data were normalized using GAPDH as an internal control, and calculations were performed using the ΔΔCt- method. The primers' Assay IDs are shown in S2 Table.

## Capillary Electrophoresis Time-of-Flight Mass Spectrometry (CE-TOFMS) measurement for metabolome analysis

**Metabolite extraction.** Metabolite extraction and metabolome analysis were conducted at Human Metabolome Technologies (HMT) (Tsuruoka, Japan). Briefly, 50 μL of serum were added to 450 μL of methanol containing internal standards (Solution ID: H3304-1002, Human Metabolome Technologies, Inc., Tsuruoka, Japan) at 0°C in order to inactivate enzymes. The extract solution was thoroughly mixed with 500 μL of chloroform and 200 μL of Milli-Q water and centrifuged at 2,300 ×$g$ and 4°C for 5 min. The 400 μL of upper aqueous layer was centrifugally filtered through a Millipore 5-kDa cutoff filter to remove proteins. The filtrate was centrifugally concentrated and re-suspended in 50 μL of Milli-Q water for metabolome analysis at HMT.

**Metabolome analysis.** A quantitative analysis of the targeted charged metabolites by CE-TOFMS was conducted as previously described [30, 31]. Briefly, CE-TOFMS analysis was carried out using an Agilent CE capillary electrophoresis system equipped with an Agilent 6210 time-of-flight mass spectrometer, Agilent 1100 isocratic HPLC pump, Agilent G1603A CE-MS adapter kit, and Agilent G1607A CE-ESI-MS sprayer kit (Agilent Technologies, Waldbronn, Germany). The systems were controlled by Agilent G2201AA ChemStation software version B.03.01 for CE (Agilent Technologies, Waldbronn, Germany) and connected by a fused silica capillary (50 μm *i.d.* × 80 cm total length) with commercial electrophoresis buffer (H3301-1001 and H3302-1021 for cation and anion analyses, respectively, HMT) as the electrolyte. The spectrometer was scanned from *m/z* 50 to 1,000 [30]. Peaks were extracted using MasterHands, automatic integration software (Keio University, Tsuruoka, Japan) in order to obtain peak information including *m/z*, peak area, and migration time (MT) [32]. Signal peaks corresponding to isotopomers, adduct ions, and other product ions of known metabolites were excluded, and remaining peaks were annotated according to the HMT metabolite database based on their *m/z* values with the MTs. Areas of the annotated peaks were then normalized based on internal standard levels and sample volumes in order to obtain relative levels of each metabolite. Hierarchical cluster analysis was performed by HMT's proprietary software, PeakStat.

## Gut microbiome analyses

**Illumina library generation.** Feces from individual subject were collected by feces collection container (TechnoSuruga Laboratory, Japan) and immediately frozen at under -25°C, and

kept until using analysis. The genomic DNA extraction of gut microbiota and microbiome analysis were conducted at TechnoSuruga Laboratory Co., Ltd. (Shimizu, Japan), as previously described by Takahashi et al. [33]. Next-generation sequencing analysis was performed using a MiSeq (Illumina, San Diego, CA). The V3-V4 regions of bacterial 16S rRNA were amplified using the 341 F (5′-CCTACGGGAGGCAGCAG-3′) [34] and 806R (5′-GGACTACHVGGG TWTCTAAT-3′) [35] primers and dual-index method [36].

**Illumina sequencing and quality filtering.**   Barcoded amplicons were paired-end sequenced on 2×284-bp cycle using the MiSeq system with MiSeq Reagent Kit version 3 (600 Cycle) chemistry. Paired-end sequencing reads were merged using fastq-join program with default settings (https://expressionanalysis.github.io/ea-utils/). Only joined-reads that had quality value score of ≥ 20 for more than 99% of the sequence were extracted using FAS-TX-Toolkit (hannonlab.cshl.edu/fastx_toolkit/). The chimeric sequences detected by usearch6.1.544_i86 [37] were deleted for further analysis.

**Bioinformatics analysis.**   The filter-passed 16S rDNA reads were subjected to homology searching using Metagenome@KIN Ver 2.2.1 analysis software (World Fusion Co., Ltd., Tokyo, Japan) and the TechnoSuruga Lab Microbial Identification Database DB-BA13.0 (TechnoSuruga Laboratory, Japan) with homology for ≥ 97% [38].

## Statistical analyses

Values are means ± SEM. Statistical significance was evaluated using GraphPad Prism, version 7 (GraphPad Software, San Diego, CA). For comparison between two groups, *Wilcoxon signed-rank test* was performed. For multiple comparisons, ANOVA with Tukey's post hoc test was performed. Statistical significance was defined as p≦0.05.

# Results

## Effects of oral administration of paramylon on biological parameters and renal function in CKD rat model

CKD rat model showed body weight loss, reduced food intake, and increased water intake (Table 1 and Fig 1A). All of these measured parameters were comparable between the Nx and Nx + PAR groups. Both increases in serum urea nitrogen, creatinine, and total cholesterol, and

**Table 1.  Effects of oral administration of paramylon on biological parameters and renal function in control and CKD rat model.**

|  | Control (n = 4) | Nx (n = 8) | Nx + PAR (n = 8) |
|---|---|---|---|
| BW(g) at 8wk | 287±13 | 263±9.4 | 271±4.3 |
| Diet intake (g/day) | 17.5±1.8 | 11±1.5* | 15±0.63 |
| Water intake (g/day) | 18±2.0 | 36.9±6.1 | 43.4±4.8* |
| Serum UN (mg/dL) | 19.0±0.84 | 64.6±10* | 43.9±2.7 |
| Serum Cr (mg/dL) | 0.33±0.0 | 1.2±0.3* | 0.73±0.1 |
| Ccr (ml/min/100g BW) | 80.7±5.3 | 22.6±4.7 * | 31.8±2.7 * |
| Serum total cholesterol (mg/dL) | 80.5±7.8 | 227±33 * | 154±17 |
| Urinary protein (mg/mg Cr) at 8wk | 4.26±0.19 | 160±49 * | 66.6±16# |

CKD, chronic kidney disease; Nx, 5/6 nephrectomy group; Nx + PAR, 5/6 nephrectomy + 5% paramylon treatment group; BW, body weight; UN, urea nitrogen; Cr, creatinine; Ccr, creatinine clearance

* P<0.05 versus Control

# P<0.05 versus Nx.

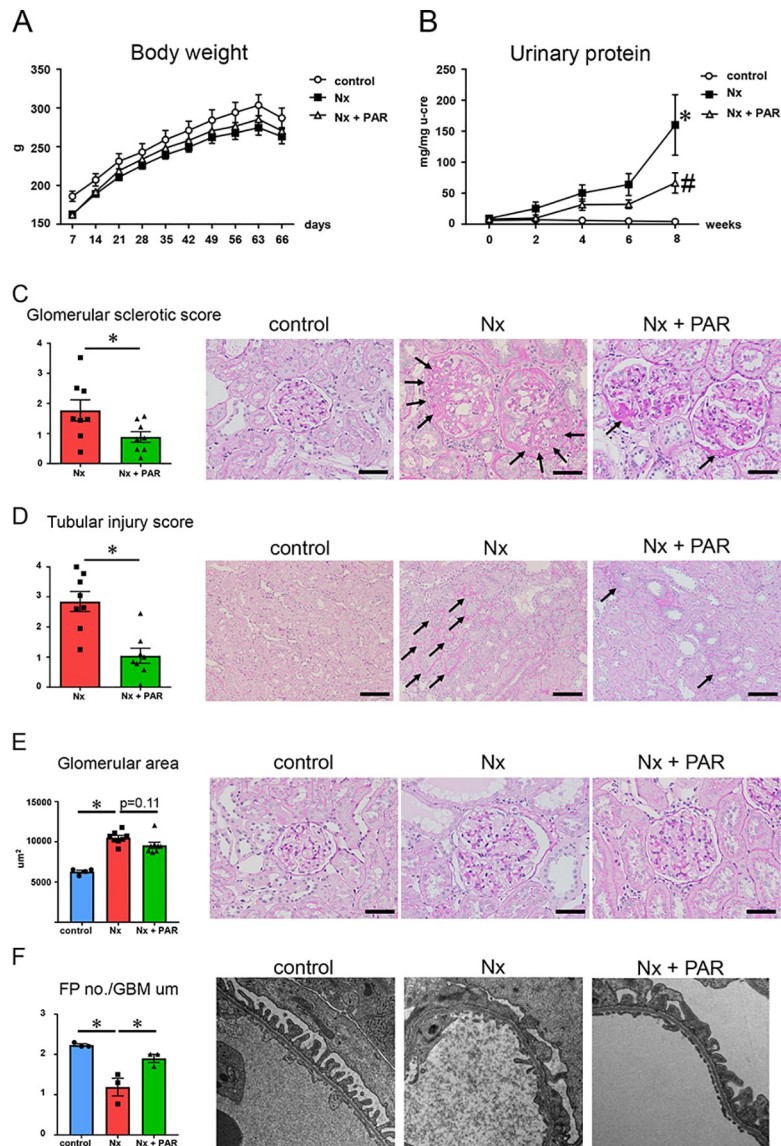

**Fig 1. Effect of paramylon on body weight, urinary protein, and renal histology in CKD rat model.** (A) Changes of body weight. (B) Changes of urinary protein. * p<0.05 vs control, # p<0.05 vs Nx; n = 8/group, except control group (n = 4). (C) Glomerular sclerotic score evaluation by periodic acid-Schiff (PAS) staining. Micrographs show the representative glomerular cross-sections. Arrows indicate sclerotic area. Scale bar = 50 μm. (D) Tubular injury score evaluation by PAS staining. Micrographs show the representative renal cortices. Arrows indicate injured tubule. Scale bar = 100 μm. (E) Glomerular hypertrophy evaluation by PAS staining. Micrographs show the representative glomerular cross-sections. Scale bar = 50 μm. *p<0.05; n = 8/group, except control group (n = 4). (F) Quantification of the number of foot processes (FP) per micrometer of glomerular basement membrane (GBM). Scale bar = 1 μm. *p<0.05; n = 3/group. CKD, chronic kidney disease; Nx, 5/6 nephrectomy group; Nx + PAR, 5/6 nephrectomy + 5% paramylon treatment group.

decreases in creatinine clearance (Ccr) in the Nx group were alleviated in the Nx + PAR group; however, there was no statistically significant difference (Table 1). An increase in urinary protein excretion in the Nx group was significantly suppressed in the Nx + PAR group at 8 weeks (Table 1 and Fig 1B). These results indicated that the renal dysfunction that existed in the Nx group was alleviated by paramylon treatment.

## Oral administration of paramylon ameliorated glomerulosclerosis, tubular injury, and glomerular podocyte injury in CKD rat model

We examined the effect of paramylon on renal histological changes in our CKD rat model. Glomerular sclerosis and tubular injury observed in the Nx group were significantly ameliorated in the Nx + PAR group (Fig 1C and 1D). Glomerular hypertrophy observed in the Nx group was inhibited in the Nx + PAR group, although there was no significant difference (Fig 1E). Ultrastructural analysis by TEM showed that the extensive foot process effacement in the Nx group was significantly recovered in the Nx + PAR group (Fig 1F).

## Effects of paramylon on renal inflammation and fibrosis in CKD rat model

Further, since the interstitial extracellular matrix is prone to accumulation in CKD and considering that several inflammatory cell-types contribute to inflammation in the interstitial fibrotic areas, we sought to evaluate renal inflammation and fibrosis by immunohistochemistry and quantitative real-time RT-PCR. The renal fibrosis observed in the Nx group was attenuated in the Nx + PAR group, although there was no significant difference (p = 0.062) (Fig 2A). Similarly, no differences were detected with the α-SMA-positive staining area and the number of tubulointerstitial ED-1-and CD3-positive cells between the Nx group and the Nx + PAR group (Fig 2B–2D). Further, the number of tubulointerstitial PCNA -positive cells in the Nx + PAR group was significantly lower than that in the Nx group (Fig 2E). Quantitative real-time RT-PCR analysis revealed a non-significant reduction in the expression of *Ccl2*, *Tnfa*, *Serpine1*, *Il-1b*, *Tgfb1*, and *Col1a1*in the kidneys of the Nx + PAR group, compared with the Nx group (Fig 3A–3F). Finally, we investigated matrix metalloproteinase 9 (MMP9) [39], interferon-γ (IFN-γ), and inducible nitric oxide synthase2 (NOS2) as proinflammatory M1 type macrophage markers, and IL-10 and IL-4 as anti-inflammatory M2 type macrophage markers [40, 41] to determine if there was modulation of macrophage profile induced by paramylon. Our results showed no differences in the gene expressions of *Mmp9*, *Ifng*, *Nos2*, *Il-10*, and *Il-4* between the Nx group and the Nx + PAR group (Fig 3G–3K). These results indicated that paramylon treatment ameliorated the progression of renal inflammation and fibrosis in the CKD rat model.

## Accumulation of Tricarboxylic Acid (TCA) cycle metabolites and uremic toxins were decreased in the serum by paramylon treatment

Given that paramylon is a dietary fiber and thus, could suppress fat and cholesterol absorption in the digestive tract [16], we examined the effect of paramylon on eliminating uremic toxins in the CKD rat model. To clarify the compounds whose serum concentrations were changed by paramylon treatment, we employed CE-TOFMS analysis. A total of targeted 407 anionic and 545 cationic compounds were analyzed. Among them, 216 compounds (80 anions and 136 cations) were detected (S3 Table). Heatmap and hierarchical clustering analyses of the identified 216 compounds are shown in S1 Fig.

**Anions.** The relative areas of 10 anionic compounds, 2-hydroxy-4-methylvaleric acid, 2-oxoglutaric acid, 5-oxoproline, cis-aconitic acid, citric acid, homovanillic acid, isocitric acid, malic acid, succinic acid, and 2-keto-glutaramic acid, were significantly decreased in the Nx + PAR group, compared to those in the Nx group. Six of them were identified as compounds that were significantly increased in the Nx group compared to the sham controls and were significantly decreased with paramylon treatment, suggesting uremic solutes that indicate the effects of paramylon. The six anionic compounds included 5-oxoproline, cis-aconitic acid, citric acid, homovanillic acid, isocitric acid, and malic acid (Fig 4A). We did not identify

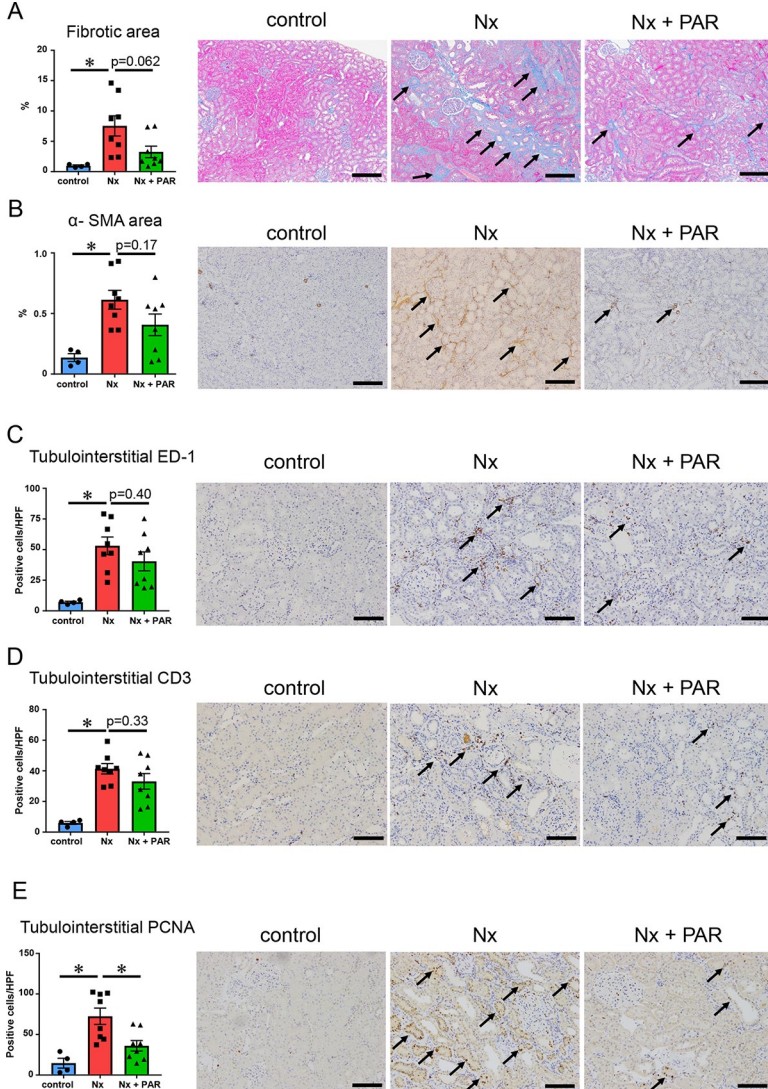

**Fig 2. Effect of paramylon on renal fibrosis and inflammation in CKD rat model.** (A) Analysis of renal fibrosis in tubulointerstitium. Micrographs show the representative renal cortices stained by Masson's trichrome. Arrows show fibrotic area. (B) Immunohistochemical analysis of renal α-smooth muscle actin (α-SMA) expression in tubulointerstitium. Micrographs show the representative renal cortices stained by α-SMA. Arrows show α-SMA positive area. (C) Immunohistochemical analysis of ED-1 positive cell numbers in tubulointerstitium. Micrographs show the representative renal cortices stained by ED-1. Arrows show ED-1 positive cells. (D) Immunohistochemical analysis of CD3 positive cell numbers in tubulointerstitium. Micrographs show the representative renal cortices stained by CD3. Arrows show CD3 positive cells. (E) Immunohistochemical analysis of proliferating cell nuclear antigen (PCNA) positive cell numbers in tubulointerstitium. Micrographs show the representative renal cortices stained by PCNA. Arrows show PCNA positive cells. CKD, chronic kidney disease; HPF, high power field; Nx, 5/6 nephrectomy group; Nx + PAR, 5/6 nephrectomy + 5% paramylon treatment group. Scale bar = 200 μm (A-B) and 100 μm (C-E). *p<0.05; n = 8/group, except control group (n = 4).

compounds that were significantly decreased in the Nx group compared to the sham controls and were significantly increased with paramylon treatment.

**Cations.** The relative areas of 34 cationic compounds, 1-methyladenosine, 1-methylhisti-dine, 1-methylnicotinamide, 4-guanidinobutyric acid, 5-hydroxylysine, asymmetric dimethyl-larginine (ADMA), alanine, arginine, argininosuccinic acid, aspartic acid, choline, citrulline, cystathionine, cytidine, ethanolamine, glutamic acid, glutathione (GSSG)_divalent,

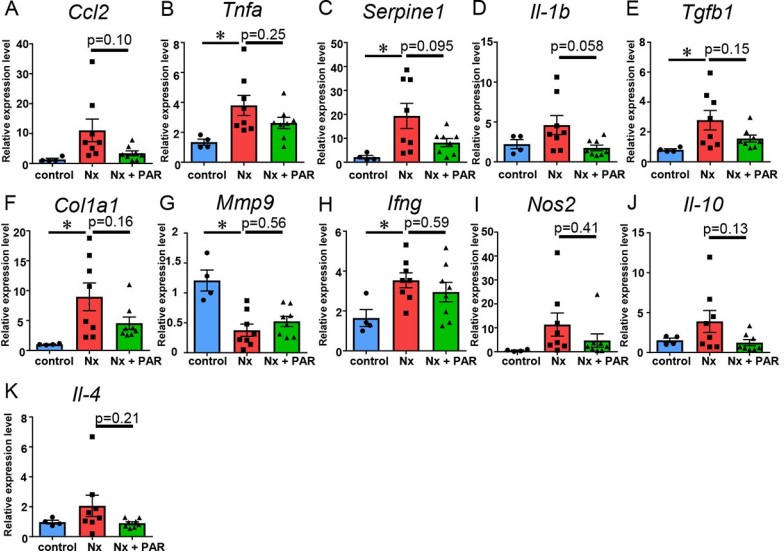

**Fig 3. Effects of paramylon on renal gene expression in CKD rat model.** Relative transcript levels of inflammatory cytokines (*Ccl2*, *Tnfa*, *Serpine1*, and *Il-1b*), fibrosis-related markers (*Tgfb1* and *Col1a1*), and markers associated with macrophage profiles (*Mmp9*, *Ifng*, and *Nos2* as proinflammatory M1 type macrophage markers, and *Il-10* and *Il-4* as anti-inflammatory M2 type macrophage markers) were measured using real-time RT-PCR. The expression levels were first normalized to those of *Gapdh* and then further normalized to the levels in the kidney from the control rats. CKD, chronic kidney disease; Nx, 5/6 nephrectomy group; Nx + PAR, 5/6 nephrectomy + 5% paramylon treatment group. $^*$ $p < 0.05$; n = 8/group, except control group (n = 4).

glycerophosphocholine, homocitrulline, methionine, N,N-dimethylglycine, $N^6,N^6,N^6$-tri-methyllysine, $N^6$-methyllysine, $N^8$-acetylspermidine, O-acetylhomoserine, phosphorylcholine, pipecolic acid, S-methylcysteine, sarcosine, symmetric dimethylarginine (SDMA), trigonelline, uracil, urea, and galactosylhydroxylysine, were significantly decreased in the Nx + PAR group, compared to those in the Nx group. Twenty-one of them were identified as compounds that were significantly increased in the Nx group compared to the sham controls and were significantly decreased with paramylon treatment, suggesting uremic solutes that indicate the effects of paramylon. The 21 cationic compounds include 1-methyladenosine, 1-methylhistidine, 1-methylnicotinamide, 4-guanidinobutyric acid, 5-hydroxylysine, ADMA, argininosuccinic acid, citrulline, cystathionine, homocitrulline, N,N-dimethylglycine, $N^6,N^6,N^6$-trimethyllysine, $N^6$-methyllysine, $N^8$-acetylspermidine, O-acetylhomoserine, pipecolic acid, sarcosine, SDMA, trigonelline, urea, and galactosylhydroxylysine (Fig 4A). We did not identify compounds that were significantly decreased in the Nx group compared to the sham controls and were significantly increased with paramylon treatment.

**TCA cycle-related metabolites.** Among the six anions suggesting uremic solutes that indicate the effects of paramylon, four anions (cis-aconitic acid, citric acid, isocitric acid, and malic acid) were TCA cycle-related metabolites. The other detected TCA cycle-related metabolites, including fumaric acid and succinic acid, were also decreased with paramylon treatment (Fig 4B).

## Effects of paramylon on gut microbiota in CKD rat model

It is well established that in CKD, some uremic toxins' precursors arising from nutrient processing by the gut microbiota produce uremic toxins, which could, in turn, alter the intestinal microbiota toward the progression of CKD. Therefore, we investigated the effect of paramylon on the gut microbiota in our CKD rat model. Next-generation sequencing revealed 26 orders,

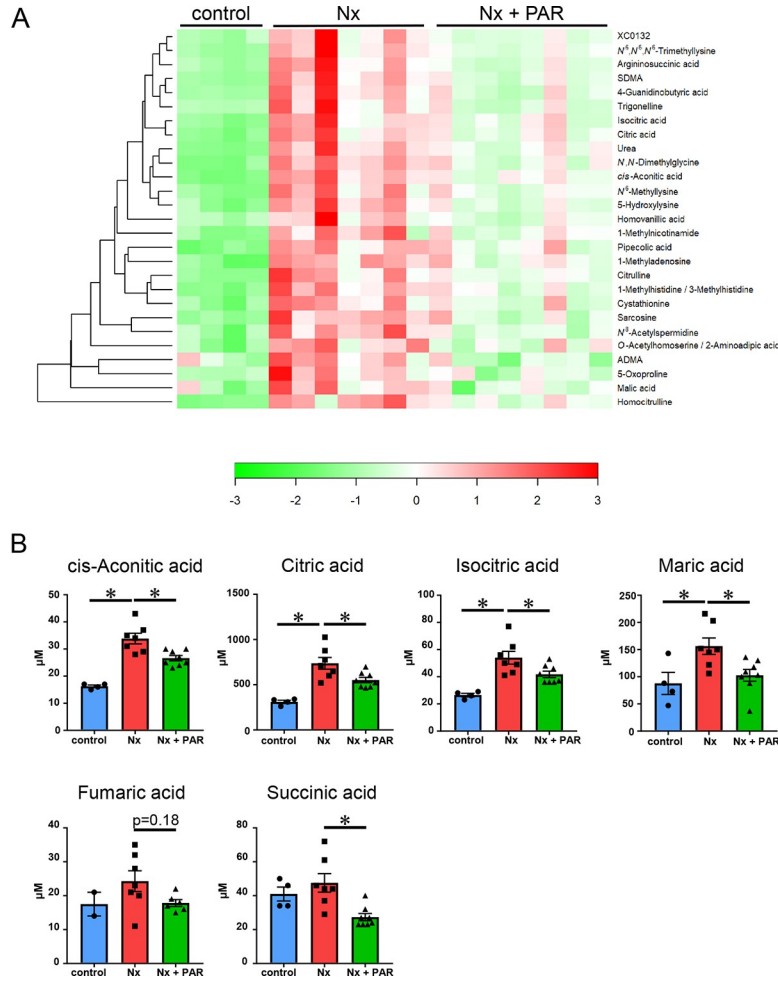

**Fig 4. Capillary Electrophoresis Time-of-Flight Mass Spectrometry (CE-TOFMS) measurement for metabolome analysis in the serum of CKD rat model.** (A) Heatmap showing six anions and 21 cations suggesting uremic solutes that indicate the effects of paramylon. Red indicates higher than average metabolite concentrations, while green indicates those below average. XC0132 denotes galactosylhydroxylysine. CKD, chronic kidney disease; ADMA, asymmetric dimethylarginine; SDMA, symmetric dimethylarginine; Nx, 5/6 nephrectomy group; Nx + PAR, 5/6 nephrectomy + 5% paramylon treatment group. control: n = 4, Nx: n = 7 (one animal was lack of serum for CE-TOFMS analysis), Nx + PAR: n = 8. (B) Increased abundance of tricarboxylic acid (TCA) cycle-related metabolites in the serum of CKD rat model were decreased by paramylon treatment. The y axis indicates the micromolar concentration. *p<0.05; control: n = 4, Nx: n = 7, Nx + PAR: n = 8 (for fumaric acid, concentrations of two control samples and two Nx + PAR group samples were not determined).

55 families, and 111 genera in fecal samples (S4–S6 Tables). The major subfamilies at the genus levels were *Lactobacillus* and *Clostridium*. *Lactobacillus* was less abundant, whereas *Clostridium* was more abundant in both the Nx group and the Nx + PAR group compared with the control group (Fig 5A). At the genus levels, it was reported that *Lactobacillus*, *Prevotella*, *Bacteroides*, *Clostridium*, *Corynebacterium*, *Anaerococcus*, *Rothia*, *Sutterella*, *Eubacterium*, *Fusobacterium*, *Leptotrichia*, *Parabacteroides*, *Peptoniphilus*, *Porphyromonas*, and *Veillonella* were associated with CKD [42, 43]. We investigated the proportional change in the representative 15 genera associated with CKD (Fig 5B). The decrease in *Lactobacillus* and *Rothia*, and the increase of *Clostridium* in the Nx group, compared with the control group, were consistent with previous studies [42, 43]. However, there were no differences in the proportions of *Bacteroides*, *Corynebacterium*, and *Parabacteroides* between the Nx group and the control group.

A

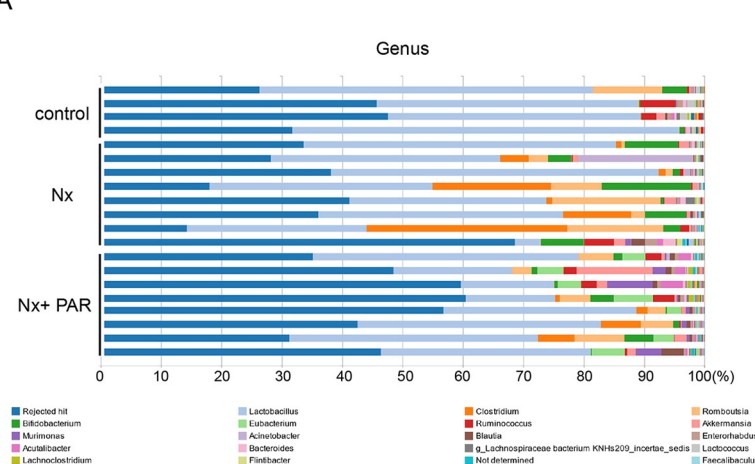

B

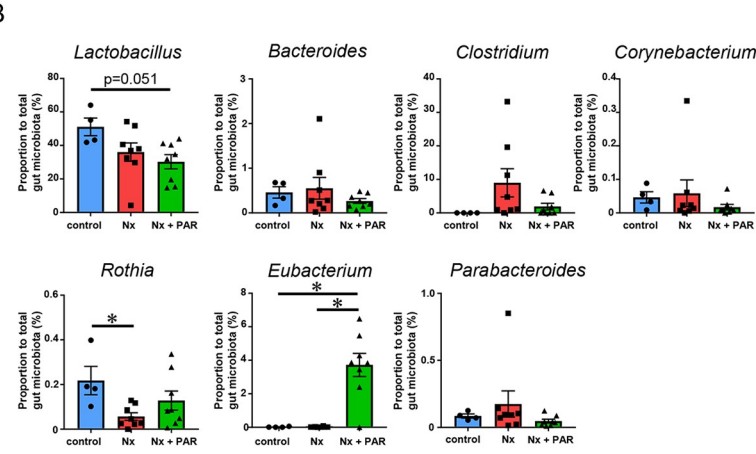

C

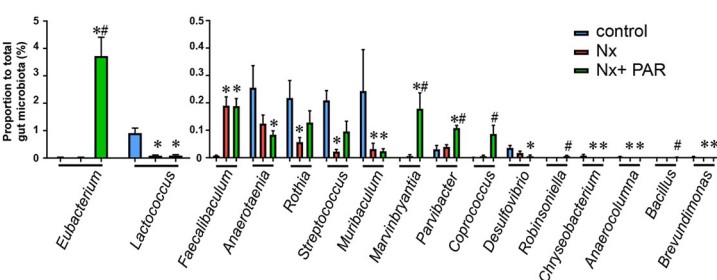

**Fig 5. Effects of paramylon on gut microbiota population in CKD rat model.** (A) Relative abundance of gut microbiota based on the average number of each subfamily at the genus levels in each sample. The representative subfamilies are indicated at the bottom. (B) The proportional change in the representative genera associated with CKD. *p<0.05 (C) The proportional change of each subfamily at the genus levels. Genera with a significant change among the three groups are shown. The figures of *Eubacterium* in (B) and (C) are identical. CKD, chronic kidney disease; Nx, 5/6 nephrectomy group; Nx + PAR, 5/6 nephrectomy + 5% paramylon treatment group. *p<0.05 vs control, # p<0.05 vs Nx; n = 8/group, except control group (n = 4).

*Eubacterium* was significantly more abundant in the Nx + PAR group compared with the Nx group. The remaining eight genera were not detected in the present study. There were 16 genera with a significant change among the three groups (Fig 5C). Of the16 genera, six genera (*Eubacterium*, *Marvinbryantia*, *Parvibacter*, *Coprococcus*, *Robinsoniella*, and *Bacillus*) were significantly more abundant in the Nx + PAR group, compared with the Nx group.

These results indicated that paramylon treatment modulated a part of the CKD-related gut microbiota in the CKD rat model.

## Discussion

In this study, we investigated if paramylon could exhibit any renoprotective effects in the CKD rat model. We showed that paramylon attenuated renal function, glomerulosclerosis, tubulointerstitial injury, and podocyte injury, leading to a decrease in proteinuria, in the CKD rat model with a remnant kidney. This finding suggests that paramylon treatment revealed clinically and histologically milder disease in this model. Although not significant, paramylon suppressed renal fibrosis and renal gene expression of proinflammatory cytokines. The number of tubulointerstitial PCNA positive cells significantly decreased by paramylon. The first major finding of this study was that the anti-inflammatory and anti-fibrotic effects of paramylon were associated with the protection of renal injury in this model.

Next, given that paramylon is a dietary fiber and thus, could suppress fat and cholesterol absorption in the digestive tract, we investigated whether paramylon inhibited absorption of the uremic toxin with CE-TOFMS analysis. As a result, we identified six anionic and 21 cationic compounds suggesting uremic solutes that indicate the effects of paramylon. Of these compounds, 20 of them were reported as uremic toxins in CKD mice or CKD patients (S7 Table). Our second major finding was that paramylon inhibited the accumulation of uremic toxins. Of the six anions we identified as uremic toxins, four of them were TCA cycle-related metabolites. It has been reported that TCA cycle-related metabolites accumulated in the kidneys of diabetic mice, which was reversed by inhibition of oxidative stress, inflammation, or fibrosis [44]. It is unclear that TCA cycle-related metabolites accumulate in the kidneys of the non-diabetic CKD rodent model; however, plasma levels of TCA cycle-related metabolites in the non-diabetic CKD mice increased (S7 Table). Anti-inflammatory and anti-fibrotic effects of paramylon may inhibit the increase of TCA cycle-related metabolites in the serum of the CKD rat model.

Third, given that various uremic solutes, including indoxyl sulfate, are derived from the microbial metabolic activities in the gut [45], we investigated the gut microbiota in our study. At the genus level, there were no differences in the proportional change of the representative CKD-related microbiota, except *Eubacterium* between the Nx group and the Nx + PAR group. However, quantitative analysis revealed that six genera, including *Eubacterium*, *Marvinbryantia*, *Parvibacter*, *Coprococcus*, *Robinsoniella*, and *Bacillus*, were significantly more abundant in the Nx + PAR group, compared with the Nx group. It was uncertain that *Robinsoniella* and *Bacillus* played a vital role in the present CKD model because the proportions of them were too low. There was the possibility that a paramylon diet *per se* induced the alteration of *Eubacterium*, *Marvinbryantia*, *Parvibacter*, and *Coprococcus*. However, the previous report has shown that *Eubacterium* was depleted in renal failure, leading to renal injury [46]. Thus, the increase in *Eubacterium* induced by paramylon may protect renal damage. Our third major finding was that paramylon modulated a part of the CKD-related gut microbiota, including *Eubacterium*. Among the uremic toxins we identified, TCA cycle-related metabolites and eight cations, including ADMA, citrulline, N6,N6,N6,-trimethyllysine, pipecolic acid, sarcosine, SDMA, trigonelline, and urea were not regarded as microbiota-derived uremic solutes [45].

Therefore, we speculated that paramylon mainly inhibited the absorption of non-microbiota-derived uremic solutes, leading to the protection of the renal injury shown in the present CKD model.

There were several weaknesses and strengths in our study. First, not all parameters studied in paramylon treatment reached statistical significance. However, there was consistency in the beneficial changes observed in the paramylon treatment group. Second, it was unknown whether the uremic toxins we identified were directly absorbed by paramylon because we did not measure these compounds in the rats' feces and tissues. On the other hand, one strength of our study is that paramylon intervention demonstrated its therapeutic potential in CKD. Therefore, our findings open an avenue for the relevance of paramylon to be utilized in human clinical trials in CKD.

In conclusion, paramylon inhibited the accumulation of uremic toxins, including TCA cycle-related metabolites. It modulated a part of the CKD-related gut microbiota in the CKD rat model with a remnant kidney. We inferred that paramylon mainly inhibited the absorption of non-microbiota-derived uremic solutes, leading to protect renal injury via anti-inflammatory and anti-fibrotic effects. Paramylon may be a novel compound that could be used against the progression of CKD. Further investigation is needed to understand the detailed functions of paramylon to better aid its plausible utilization in the prevention of CKD progression.

## Supporting information

**S1 Fig. Heatmap and hierarchical clustering analyses of the detected 216 compounds in the serum.** Red indicates higher than average metabolite concentrations, while green indicates those below average. Nx, 5/6 nephrectomy group; Nx + PAR, 5/6 nephrectomy + 5% paramylon treatment group.
(TIF)

**S1 Table. Primary antibodies.**
(DOCX)

**S2 Table. TaqMan® Gene Expression Assay IDs.**
(DOCX)

**S3 Table. Dataset for CE-TOFMS analysis.** KEGG ID is the ID registered with KEGG database (http://www.genome.jp/kegg/). HMDB ID is the ID registered with Human Metabolome Project database (http://www.hmdb.ca/).
(DOCX)

**S4 Table. Dataset for taxon counts at the order levels in the fecal samples.**
(DOCX)

**S5 Table. Dataset for taxon counts at the family levels in the fecal samples.**
(DOCX)

**S6 Table. Dataset for taxon counts at the genus levels in the fecal samples.**
(DOCX)

**S7 Table. Previous reports as uremic toxin of six anionic and 21 cationic compounds we identified in the present study.**
(DOCX)

**S8 Table. Dataset for biological parameters and renal function.**
(DOCX)

**S9 Table. Dataset for body weight and urinary protein.**
(DOCX)

**S10 Table. Dataset for renal histology by periodic acid-Schiff staining and electron microscopy.**
(DOCX)

**S11 Table. Dataset for analysis of renal fibrosis by Masson's trichrome staining and immunohistochemical analysis of renal α-Smooth Muscle Actin (α-SMA) expression, and ED-1, CD3, and Proliferating Cell Nuclear Antigen (PCNA) positive cell numbers in tubulointerstitium.**
(DOCX)

**S12 Table. Dataset for relative transcript levels of *Ccl2*, *Tnfa*, *Serpine1*, *Il-1b*, *Tgfb1*, *Col1a1*, *Mmp9*, *Ifng*, *Nos2*, *Il-10*, and *Il-4* in kidney by real-time RT-PCR.**
(DOCX)

**S13 Table. Dataset for serum concentrations of TCA cycle-related metabolites.**
(DOCX)

## Acknowledgments

The authors thank Yuko Arakawa, Masahito Yamada, Kazutoshi Unno, and Kazuhiko Yuzawa for their technical assistances.

## Author Contributions

**Formal analysis:** Yoshikuni Nagayama, Naoyuki Isoo, Mizuki Yamano, Tomoyuki Nariyama, Motoka Yagame.

**Investigation:** Yoshikuni Nagayama, Naoyuki Isoo, Ayaka Nakashima, Kengo Suzuki, Mizuki Yamano, Tomoyuki Nariyama, Motoka Yagame.

**Project administration:** Yoshikuni Nagayama.

**Resources:** Ayaka Nakashima, Kengo Suzuki.

**Supervision:** Katsuyuki Matsui.

**Writing – original draft:** Yoshikuni Nagayama, Katsuyuki Matsui.

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
