## [Decision Letter · Decision Letter 0]

13 Feb 2020

PONE-D-19-35917

Renoprotective effects of paramylon, a β-1,3-D-Glucan isolated from Euglena gracilis Z in a rodent model of chronic kidney disease

PLOS ONE

Dear Dr. Nagayama,

Thank you for submitting your manuscript to PLOS ONE. After careful consideration, we feel that it has merit but does not fully meet PLOS ONE’s publication criteria as it currently stands. Therefore, we invite you to submit a revised version of the manuscript that addresses the points raised during the review process.

The manuscript was revised and we appreciate if the authors could reply to all issues discriminated below.

We would appreciate receiving your revised manuscript by Mar 29 2020 11:59PM. To enhance the reproducibility of your results, we recommend that if applicable you deposit your laboratory protocols in protocols.io, where a protocol can be assigned its own identifier (DOI) such that it can be cited independently in the future. For instructions see: http://journals.plos.org/plosone/s/submission-guidelines#loc-laboratory-protocols

We look forward to receiving your revised manuscript.

Kind regards,

Niels Olsen Saraiva Câmara, M.D, PhD

Academic Editor

PLOS ONE

Journal Requirements:

2. Thank you for stating the following in the Competing Interests/Financial Disclosure* (delete as necessary) section:

"I have read the journal's policy and the authors of this manuscript have the following competing interests: Ms. Ayaka Nakashima and Dr. Kengo Suzuki are employees of euglena Co., Ltd."

We note that one or more of the authors are employed by a commercial company: euglena Co., Ltd.

Reviewers' comments:

Reviewer's Responses to Questions

**Comments to the Author**

1. Is the manuscript technically sound, and do the data support the conclusions?

Reviewer #1: Yes

Reviewer #2: Partly

2. Has the statistical analysis been performed appropriately and rigorously? 

Reviewer #1: I Don't Know

Reviewer #2: Yes

3. Have the authors made all data underlying the findings in their manuscript fully available?

Reviewer #1: Yes

Reviewer #2: Yes

4. Is the manuscript presented in an intelligible fashion and written in standard English?

Reviewer #1: Yes

Reviewer #2: Yes

5. Review Comments to the Author

Reviewer #1: The manuscript submitted by Nagayama et al describes the renoprotective effects of paromylon, a Beta-1,3-D-Glucan in a rodent model of chronic kidney disease. The treatment showed improvement of the kidney function and decreased accumulation of TCA cycle metabolites and uremic toxins.

Major comments:

-Figure 2 a, b and C: use arrows to show where the injury/damage is on the pictures for glomerulosclerosis, tubular injury, glomerular hypertrophy.

- Improve the images quality. Show the bar graphs with dots for each mouse.

- Figure 1 can be joint to Figure 2, as well as the results sections linked to both figures.

-Figure 3: Improve the images quality. Use arrows to show where the injury/damage is.

Don't use the word tendency to describe the non-significant differences in the results section. Use dots to represent each mouse on the graphs (dot plot).

- Figure 4: It would be nice to see MMP9, IL10 and ROS levels in the kidneys too. It would be interesting to evaluate if there is modulation of the macrophage profile induced by paromylon.

-Figure 5: The treatment decreases the accumulation of TCA cycle metabolites. Improve the discussion about these data. Correlate to improvement of inflammatory parameters such as IL1b.

Improve the quality of the graphs. An alternative to better show this data is to build a heatmap.

The results section “Identification of some uremic toxins using CE-TOFMS analysis” needs to be improved. The title could be changed to something like: Accumulation of TCA cycle metabolites and uremic toxins are decreased in the serum by paromylon treatment.

This section should be better described because these results fall from the sky, without any introduction about why to look at this (TCA cycle etc).

Figure 6: Since Beta glucans are found in fungi species, would you consider changes in fungi species in the microbiome induced by the treatment?

Minor Comments:

-The word “tendency” should be replaced.

-Improve the quality of all images and include the scale bar.

Reviewer #2: This is an interesting manuscript which authors indicating paramylon reduces renal tissue damage and inflammation, however, paramylon has no effect on the intestinal microbiota. However, several points to be revised in the present manuscript to be accept in Plos One, described as follow.

Introduction:

- Must be revised and include an information about the impact of altered intestinal microbiota on CKD progression and the beneficial of dietary fibers in this context.

- Must be cited the original studies about the biological effects of dietary B-glucans described in lines 58-60.

- After citing once “Euglena gracilis Z”, abreviate as “E. gracilis Z”, as described in the others manuscript’s sections.

- Instead of use “CKD rats”, substitute for “CKD rat model”.

Results:

- Revised the use of the “on the other hand” in description of results.

- The scale bars must be insert in the images.

- In the figure 4, identify de graphics with letters (A-F), the control group was not demonstrated in this figure.

- Attenuate the phrase described in lines 306-310. The authors just analyzed gene expression.

- The authors reported no differences in bacteria genera or clusters in microbiota. Maybe the differences could be observed among the species, instead of genera or clusters. Metagenomic assay will be very much appreciated to verify this point.

- In the literature has been hypothesized that increased gut permeability contributes to inflammation in patients with advanced CKD. Was the gut tissue analyzed in the present study? Sometimes the amelioration induced by paramylon could be related to improve de mucosal permeability, and it could be discussed.

Discussion

-The results are very poor discussed. The authors described results in this section instead of discuss them. The discussion must be revised and intersect the findings with data from literature.

-The limitations of techniques used to characterize intestinal microbiome must be discussed. The interface between gut microbiota and CKD must be explored.

In line 452 - “It is unknown whether 5-hydroxylysine, argininosuccinic acid,

N6-methyllysine, N8-acetylspermidine, O-acetylhomoserine, and

galactosylhydroxylysine are uremic toxins”. Were these compounds described in CKD patients or another rodent model of CKD? What the relevance these compounds whether they are unknown their classification as uremic toxins?

6. PLOS authors have the option to publish the peer review history of their article (what does this mean?). If published, this will include your full peer review and any attached files.

Reviewer #1: No

Reviewer #2: No

---

## [Author Response · Author response to Decision Letter 0]

13 May 2020

We have addressed all the comments by two reviewers, as indicated on the attached document

Funding statement: Ms. Ayaka Nakashima and Dr. Kengo Suzuki are employees of euglena Co., Ltd. They adjusted the test concept, created a sample for this experiment (preparing resources), and discussed the results obtained. This research did not receive any specific grant from funding agencies in the public, commercial, or not-for-profit sectors.

 Competing Interests Statement: Ms. Ayaka Nakashima and Dr. Kengo Suzuki are employees of euglena Co., Ltd. There are no patents, products in development or marketed products to declare. This does not alter the authors’ adherence to all the PLoS ONE policies on sharing data and materials.

---

## [Decision Letter · Decision Letter 1]

8 Jun 2020

PONE-D-19-35917R1

Renoprotective effects of paramylon, a β-1,3-D-Glucan isolated from Euglena gracilis Z in a rodent model of chronic kidney disease

PLOS ONE

Dear Dr. Nagayama,

Thank you for submitting your manuscript to PLOS ONE. After careful consideration, we feel that it has merit but does not fully meet PLOS ONE’s publication criteria as it currently stands. Therefore, we invite you to submit a revised version of the manuscript that addresses the points raised during the review process.

Specifically the authors could pay special attention to results section.

We look forward to receiving your revised manuscript.

Kind regards,

Niels Olsen Saraiva Câmara, M.D, PhD

Academic Editor

PLOS ONE

Reviewers' comments:

Reviewer's Responses to Questions

**Comments to the Author**

1. If the authors have adequately addressed your comments raised in a previous round of review and you feel that this manuscript is now acceptable for publication, you may indicate that here to bypass the “Comments to the Author” section, enter your conflict of interest statement in the “Confidential to Editor” section, and submit your "Accept" recommendation.

Reviewer #1: All comments have been addressed

Reviewer #2: (No Response)

2. Is the manuscript technically sound, and do the data support the conclusions?

Reviewer #1: Yes

Reviewer #2: Yes

3. Has the statistical analysis been performed appropriately and rigorously? 

Reviewer #1: Yes

Reviewer #2: I Don't Know

4. Have the authors made all data underlying the findings in their manuscript fully available?

Reviewer #1: Yes

Reviewer #2: Yes

5. Is the manuscript presented in an intelligible fashion and written in standard English?

Reviewer #1: Yes

Reviewer #2: No

6. Review Comments to the Author

Reviewer #1: The authors provided answer for all my points. The quality of this manuscript improved considerably, however in the results section, the results could be more explored and the authors could provide more background on why they performed those experiments, before presenting the results.

Reviewer #2: The findings reported by Nagayama and colleagues are interesting and present potential to be published in Plos One. Although, the authors had accepted all suggestions made by reviewers in the first moment, the article still need to be adequate to formal language, and the revision by a native English speaker is mandatory. In this second revision, the discussion section needs more attention, the authors use inadequate grammar and unnecessary sentences to justify the findings, some examples are listed below.

Discussion

- - Table 2 is suggested as supplemental data;

- - The sentences need to be better connected, using adequate English grammar

- - In line 507, “we investigated whether if paramylon inhibited absorption of uremic toxin with CE-TOFMS analysis”. Review the use of whether/if

- - Some phrases depreciated this work and must be attenuated or excluded: “Unfortunately, we did not get the gut tissue to analyze”; There were several limitations in our study.

- - Dear authors, the discussion must be carefully revised and interconnect the results and then submitted to a native English speaker.

7. PLOS authors have the option to publish the peer review history of their article (what does this mean?). If published, this will include your full peer review and any attached files.

Reviewer #1: No

Reviewer #2: No

---

## [Author Response · Author response to Decision Letter 1]

1 Jul 2020

PLOS ONE Manuscript ID: PONE-D-19-35917R1 

COMMENTS FOR THE REVIEWERS:

We are grateful to reviewer #1 for the critical comments and useful suggestions that have helped us to improve our paper. As indicated in the responses that follow, we have taken all these comments and suggestions into account in the revised version of our paper. Please note: all page directions apply to the manuscript in red ink version ‘Revised Manuscript with Track Changes’ (length 48 pages)

Reviewer #1: The authors provided answer for all my points. The quality of this manuscript improved considerably, however in the results section, the results could be more explored and the authors could provide more background on why they performed those experiments, before presenting the results.

 We added background on why we performed those experiments and interpretation of the results in the results section. (page 15, line 263-264, page 16, line 274, page 18, line 298-301, page 19, line 316-318, page 20, line 349-350, page 21, line 351- 353, page 24, line 412-415, page 25, line 434-435)

We are grateful to reviewer #2 for the critical comments and useful suggestions that have helped us to improve our paper. As indicated in the responses that follow, we have taken all these comments and suggestions into account in the revised version of our paper. Please note: all page directions apply to the manuscript in red ink version ‘Revised Manuscript with Track Changes’ (length 48 pages)

Reviewer #2: The findings reported by Nagayama and colleagues are interesting and present potential to be published in Plos One. Although, the authors had accepted all suggestions made by reviewers in the first moment, the article still need to be adequate to formal language, and the revision by a native English speaker is mandatory. In this second revision, the discussion section needs more attention, the authors use inadequate grammar and unnecessary sentences to justify the findings, some examples are listed below.

Discussion

- - Table 2 is suggested as supplemental data;

We replaced Table 2 with supplemental Table 7. (page 27, line 476, page 30, line 490) Because of this, we rearranged references. 

- - The sentences need to be better connected, using adequate English grammar

We submitted our manuscript to a native English speaker and we corrected the manuscript.

- - In line 507, “we investigated whether if paramylon inhibited absorption of uremic toxin with CE-TOFMS analysis”. Review the use of whether/if

We deleted ‘if’. (page 27, line 472)

- - Some phrases depreciated this work and must be attenuated or excluded: “Unfortunately, we did not get the gut tissue to analyze”; There were several limitations in our study.

We deleted the following descriptions; ‘In the literature it has been hypothesized that increased gut permeability contributes to inflammation in the advanced CKD [21]. The anti-inflammatory effect of paramylon could be related to improve the gut permeability. Unfortunately, we did not get the gut tissue to analyze.’ (page 31, line 511-514)

We deleted the following sentence; ‘There were several limitations in our study.’ (page 32, line 520) and replaced it to the attenuated phrases.

- - Dear authors, the discussion must be carefully revised and interconnect the results and then submitted to a native English speaker.

 We carefully revised the discussion section interconnected with the results and we submitted our manuscript to a native English speaker.

---

## [Decision Letter · Decision Letter 2]

21 Jul 2020

Renoprotective effects of paramylon, a β-1,3-D-Glucan isolated from Euglena gracilis Z in a rodent model of chronic kidney disease

PONE-D-19-35917R2

Dear Dr. Nagayama,

We’re pleased to inform you that your manuscript has been judged scientifically suitable for publication and will be formally accepted for publication once it meets all outstanding technical requirements.

Kind regards,

Niels Olsen Saraiva Câmara, M.D, PhD

Academic Editor

PLOS ONE

Additional Editor Comments (optional):

Reviewers' comments:

Reviewer's Responses to Questions

**Comments to the Author**

1. If the authors have adequately addressed your comments raised in a previous round of review and you feel that this manuscript is now acceptable for publication, you may indicate that here to bypass the “Comments to the Author” section, enter your conflict of interest statement in the “Confidential to Editor” section, and submit your "Accept" recommendation.

Reviewer #1: All comments have been addressed

Reviewer #2: (No Response)

2. Is the manuscript technically sound, and do the data support the conclusions?

Reviewer #1: Yes

Reviewer #2: Yes

3. Has the statistical analysis been performed appropriately and rigorously? 

Reviewer #1: Yes

Reviewer #2: I Don't Know

4. Have the authors made all data underlying the findings in their manuscript fully available?

Reviewer #1: Yes

Reviewer #2: Yes

5. Is the manuscript presented in an intelligible fashion and written in standard English?

Reviewer #1: Yes

Reviewer #2: Yes

6. Review Comments to the Author

Reviewer #1: The authors have addressed my concerns. The results section was significantly improved to better explain the findings.

Reviewer #2: (No Response)

7. PLOS authors have the option to publish the peer review history of their article (what does this mean?). If published, this will include your full peer review and any attached files.

Reviewer #1: No

Reviewer #2: No

---

## [Editor Report · Acceptance letter]

23 Jul 2020

PONE-D-19-35917R2 

Renoprotective effects of paramylon, a β-1,3-D-Glucan isolated from *Euglena gracilis* Z in a rodent model of chronic kidney disease 

Dear Dr. Nagayama:

I'm pleased to inform you that your manuscript has been deemed suitable for publication in PLOS ONE. Congratulations! Your manuscript is now with our production department. 

Kind regards, 

on behalf of

Prof. Niels Olsen Saraiva Câmara 

Academic Editor

PLOS ONE